# Enhanced Kidney Damage in Individuals with Diabetes Who Are Chronically Exposed to Cadmium and Lead: The Emergent Role for β_2_-Microglobulin

**DOI:** 10.3390/ijms26189208

**Published:** 2025-09-20

**Authors:** Soisungwan Satarug, David A. Vesey, Donrawee Waeyeng, Tanaporn Khamphaya, Supabhorn Yimthiang

**Affiliations:** 1Centre for Kidney Disease Research, Translational Research Institute, Woolloongabba, Brisbane, QLD 4102, Australia; david.vesey@health.qld.gov.au; 2Department of Kidney and Transplant Services, Princess Alexandra Hospital, Brisbane, QLD 4102, Australia; 3Environmental Health and Technology, School of Public Health, Walailak University, Nakhon Si Thammarat 80160, Thailand; donrawee.wae@wu.ac.th; 4Occupational Health and Safety, School of Public Health, Walailak University, Nakhon Si Thammarat 80160, Thailand; tanaporn.kh@mail.wu.ac.th (T.K.); ksupapor@mail.wu.ac.th (S.Y.)

**Keywords:** cadmium, diabetic kidney disease, hyperglycemia, hypertension, lead, serum β_2_-microglobulin

## Abstract

Elevated levels of circulating β_2_-microglobulin (β_2_M) are linked to an increased risk of hypertension and mortality from diabetes. The present study tests the hypothesis that the environmental pollutants, cadmium (Cd) and lead (Pb), by increasing plasma β_2_M levels, promote the development of hypertension and progression of diabetic kidney disease. Herein, we analyzed data from a Thai cohort of 72 individuals with diabetes and 65 controls without diabetes who were chronically exposed to low levels of Cd and Pb. In all subjects, serum concentrations of β_2_M inversely correlated with the estimated glomerular filtration rate (eGFR) (*r* = −0.265) and directly with age (*r* = 0.200), fasting plasma glucose (*r* = 0.210), and systolic blood pressure (*r* = 0.229). The prevalence odds ratio (POR) for hyperglycemia increased 7.7% for every 1-year increase in age and increased 3.9-fold, 3.1-fold, and 3.7-fold in those with serum β_2_M levels ≥ 5 mg/L, Cd/Pb exposure categories 2 and 3, respectively. The POR for hypertension increased 2.9-fold, 3-fold, and 4-fold by hyperglycemia (*p* = 0.011), Cd/Pb exposure categories 2 and 3. The POR for albuminuria increased 3.5-fold by hyperglycemia. In conclusion, kidney damage, evident from albuminuria, was particularly pronounced in participants with diabetes who had a serum β_2_M above 5 mg/L plus chronic exposure to low-dose Cd and Pb. For the first time, through a mediation analysis, we provide evidence that links Cd exposure to the *SH2B3*-β_2_M pathway of blood pressure homeostasis in people with and without diabetes.

## 1. Introduction

Hypertension, defined as systolic blood pressure (SBP) and/or diastolic blood pressure (DBP) ≥ 140/90 mm Hg, affects approximately one-third of the adult population in most economically developed countries [1,2]. Resistance hypertension, where a regular anti-hypertensive medication formulation fails to achieve target blood pressure control, can be found in 10-15% of those with primary or essential hypertension [3,4].

Epidemiological data linking the risk for kidney injury and hypertension to the environmental pollutants cadmium (Cd) and lead (Pb) are abundant [5,6,7,8], given that hypertension is known to be both a cause and a consequence of kidney damage [9,10,11,12,13,14]. A dose–response relationship between Cd exposure and hypertension risk has been reported in a meta-analysis by Verzelloni et al. [15]. Hypertension and diabetes are leading causes of CKD [16,17,18]. Surprisingly, however, research studies into the kidney effects of Cd and Pb exposure in people with diabetes are limited [19,20,21], as summarized below.

In a Dutch prospective cohort study, patients with diabetes who were exposed to Cd had an estimated glomerular filtration rate (eGFR) falling at a high rate [19]. A cross-sectional study on the U.S. population observed that Pb exposure may have increased the risk of CKD, especially in women with a body mass index (BMI) higher than 25 kg/m^2^ plus diabetes and who were non-smokers [20]. In a cohort of Swedish women aged 64 years, an increased risk of elevated albumin excretion (15 mg/12 h) was found only in those with diabetes who had blood Cd within the top quartile [21]. A study from Taiwan observed that Cd-related kidney damage was more extensive in women than men [22].

Excretion of β_2_-microglobulin (β_2_M) has been widely used as a marker of kidney tubular dysfunction. Recent studies, however, have linked circulating β_2_M to the *SH2B3*-β_2_M axis of blood pressure regulation [23]. The present study aimed to investigate the mechanisms of kidney damage in people with diabetes who were chronically exposed to low-dose Cd and Pb, emphasizing the role of serum β_2_M. Also, it explored how Cd/Pb exposure induces kidney tubular cell toxicity and accelerates diabetic kidney disease (DKD). In our previous case–control study [24], we reported that people chronically exposed to Cd and Pb have enhanced risks of hyperglycemia, eGFR reduction, and albuminuria. It is likely that one or both metals cause these adverse outcomes.

Herein, we hypothesize that Cd/Pb exposure and/or diabetes (hyperglycemia) increase plasma β_2_M levels, which, in turn, raises blood pressure and induces kidney tubular cell damage.

## 2. Results

### 2.1. Description of Participants

Participants were assigned according to the tertile of serum β_2_M levels (Table 1).

As shown in Table 1, participants were predominantly women (78.1%), which distributed equally across the [β_2_M]_s_ tertiles. The highest proportion of participants with diagnosed diabetes (72%), hypertension (69%), albuminuria (39%), fasting plasma glucose (FPG) ≥ 110 mg/dL (69.6%), and FPG ≥ 126 mg/dL (58.7%) were found in the [β_2_M]_s_ top tertile. The parameters showing significant variations across the [β_2_M]_s_ tertiles were SBP, eGFR, FPG, blood Cd, and blood Pb concentrations. The variations in age, BMI, DBP, E_Cd_/E_cr_, E_Cd_/C_cr_, E_alb_/E_cr_ (ACR), and E_alb_/C_cr_ across the [β_2_M]_s_ tertiles did not differ statistically, as did the distribution of those with a low eGFR.

### 2.2. Bivariate Correlation Analysis

The relationships of serum β_2_M with other variables were assessed with Spearman’s rank correlation analysis. Ten variables tested were age, BMI, FPG, SBP/DBP, eGFR (a clinical kidney function measure), E_alb_/C_cr_ (a sign of kidney damage and diabetic kidney disease), E_β2M_/C_cr_ (an indicator of tubular dysfunction), and toxic metal exposure, reflected by E_Cd_/C_cr_, and blood Cd and Pb levels. Results are tabulated (Table 2).

Serum β_2_M concentration ([β_2_M]_s_) varied directly with age (*r* = 0.200), FPG (*r* = 0.210), SBP (*r* = 0.229), and E_β2M_/C_cr_ (*r* = 0.390), while showing an inverse correlation with eGFR (*r* = −0.265). The strength of all five correlations was moderate. The correlation between [β_2_M]_s_ with all other variables, BMI, DBP, E_alb_/C_cr_, E_Cd_/C_cr_, and Cd/Pb exposure levels, were statistically insignificant.

Interestingly, the FPG varied directly with SBP (*r* = 0.250), E_alb_/C_cr_ (*r* = 0.273), E_β2M_/C_cr_ (*r* = 0.306), and Cd/Pb exposure levels (*r* = 0.181). Other notable correlations were SBP vs. E_alb_/C_cr_ (*r* = 0.372), SBP vs. E_β2M_/C_cr_ (*r* = 0.237), eGFR vs. E_Cd_/C_cr_ (*r* = −0.227), eGFR vs. E_β2M_/C_cr_ (*r* = −0.515), and E_Cd_/C_cr_ vs. Cd/Pb exposure categories (*r* = 0.301).

Because a reduction in the eGFR to 60 mL/min/1.73 m^2^ or below signifies chronic kidney disease (CKD), and because the eGFR falls as blood pressure rises, the relationships of [β_2_M]_s_ with SBP were examined further in subgroups using scatterplots and a regression analysis. Results are presented in Figure 1.

An association of SBP with [β_2_M]_s_ was significant in the eGFR subgroups (Figure 1a). The SBP and [β_2_M]_s_ association was especially strong in those with eGFR levels commensurate to CKD (R^2^ = 0.252). The association of SBP and [β_2_M]_s_ in the normal eGFR group was also statistically significant, although the R^2^ value was small (0.039). However, unlike SBP, [β_2_M]_s_ was associated in DBP only in the low eGFR group (R^2^ = 0.312) (Figure 1b).

Based on bivariate correlation data (Table 2), a simple mediation model was used to examine the indirect effects of Cd on [β_2_M]_s_ through kidney tubular toxicity, reflected by the excretion rate of β_2_M (E_β2M_/C_cr_) (Figure 2).

The Sobel test of (a*b) indicated that the effect of Cd on serum β_2_M levels was through Eβ_2_M, while its direct effect did not reach a statistically significant level. The mediating effects of Cd on SBP/DBP were examined also, as shown in Figure 3.

Cd indirectly influenced the SBP (Figure 3a) but not DBP (Figure 3b). The direct effects of Cd on SBP and DBP were statistically insignificant.

Because E_alb_/C_cr_, like E_β2M_/C_cr_, can reflect an impairment in tubular function, the correlation of this parameter with SBP was examined further in subgroups by scatterplots and a regression analysis (Figure 4).

E_alb_/C_cr_ was associated with SBP only in those with diagnosed diabetes (Figure 4a). Similarly, E_alb_/C_cr_ was associated with SBP only in those with FPG levels ≥ 126 mg/dL (Figure 4b). However, the E_alb_/C_cr_ and DBP association was not significant in any subgroup (Figure 4c,d).

### 2.3. Logistic Regresssion Model for Serum β_2_M Higher than the Median 5 mg/L

Variables that influenced serum β_2_M levels are listed in Table 3.

The prevalence odds ratio (POR) for a high serum β_2_M was minimally affected by age, BMI, E_Cd_/C_cr_, gender, smoking, and hypertension. It was affected by diagnosed diabetes and eGFR. The POR for a high serum β_2_M rose 4-fold (*p* = 0.001) in participants with diagnosed diabetes. A 4% decrease in the POR for the high serum β_2_M was associated with a 1 mL/min/1.73 m^2^ higher eGFR (*p* = 0.005).

### 2.4. Logistic Regresssion Model for Hyperglycemia

To examine the potential association between [β_2_M]_s_ ≥ 5 mg/L and an elevation of FPG, logistic regression models were undertaken for FPG ≥ 110 mg/dL and ≥ 126 mg/dL (Table 4).

The POR for prediabetes (FPG ≥ 110 mg/dL) rose 3.4-fold (*p* = 0.002) and 2.8-fold (*p* = 0.004) in those with a high serum β_2_M and Cd/Pb exposure category 3, respectively. The POR for diabetes (FPG ≥ 126 mg/dL) rose with age, a high serum β_2_M (POR 3.8, *p* = 0.002), Cd/Pb exposure category 2 (POR 3.1, *p* = 0.021), and category 3 (POR 3.7, *p* = 0.014). The POR for FPG ≥ 126 mg/dL increased 7.7% for every 1-year increase in age (*p* = 0.004).

### 2.5. Logistic Regresssion Model for Hypertension and Albuminuria

In the bivariate analysis (Table 2), the FPG varied directly with SBP (*r* = 0.250), E_alb_/C_cr_ (*r* = 0.273), E_β2M_/C_cr_ (*r* = 0.306), and Cd/Pb exposure categories (*r* = 0.181). To further explore the interrelationships, additional logistic regression models were used (Table 5 and Table 6).

As shown in Table 5, age, BMI, and gender had little effect on the POR for hypertension and albuminuria. The POR for hypertension was increased 7-fold in non-smokers (*p* = 0.020), 3.7-fold in those with FPG ≥ 110 mg/dL (*p* = 0.001), 3.1-fold and 4.4-fold in those within the Cd/Pb exposure category 2 (*p* = 0.046) and category 3 (*p* = 0.005). In comparison, the POR for albuminuria increased 2.95-fold in those with FPG ≥ 110 mg/dL (*p* = 0.013) but was not affected by the other six variables.

As shown in Table 6, the POR for hypertension rose 8-fold in non-smokers (*p* = 0.018), while it rose 2.9-fold, 3-fold, and 4-fold in those with FPG ≥ 126 mg/dL (*p* = 0.011) and in the Cd/Pb exposure category 2 (*p* = 0.050) and category 3 (*p* = 0.008). The POR for albumin rose 3.5-fold in those with FPG ≥ 126 mg/dL (*p* = 0.005).

The scatterplots and the regression analysis for E_alb_/C_cr_ vs. FPG and E_alb_/C_cr_ vs. blood pressure are presented in Figure 3.

E_alb_/C_cr_ was associated with FPG but only in participants with hypertension (Figure 5a). The E_alb_/C_cr_ and FPG association in the low eGFR group was much tighter compared to the normal eGFR group (R^2^ 0.303 vs. 0.065) (Figure 5b). E_alb_/C_cr_ in those with high serum β_2_M varied more closely with SBP than those with low serum β_2_M (R^2^ 0.155 vs. 0.099) (Figure 5c). E_alb_/C_cr_ was associated with DBP but only in the high serum β_2_M group (R^2^ = 0.103) (Figure 5d).

## 3. Discussion

### 3.1. Hypertension Associated with Hyperglycemia and Environmental Cd and Pb

Consistent with numerous literature reports, the risk of hypertension among participants was increased by the exposure to low doses of Cd and Pb (Table 5). Such low environmental exposure to Cd and Pb was suggested by mean values for blood Pb (4.49 mg/dL), blood Cd (0.57 µg/L), urinary Cd (0.65 µg/L), and the Cd excretion rate (0.98 µg/g creatinine) (Table 1). These low exposure levels were comparable to those found in most environmentally exposed populations, reported in a recent dose–response meta-analysis [15]. Concerningly, the risk of resistance hypertension was increased 30–35% by Cd exposure in the representative U.S. population, assessed with blood Cd levels [9]. In another U.S. population study (NHANES 2005–2016), urinary Cd ranging between 0.025 and 0.501 µg/L correlated with diabetes [25]. The risk of hypertension in a Korean population increased 29, 47, and 78% by exposure to Pb and Cd alone and Cd plus Pb, respectively, [13].

Previously, rising SBP after Cd exposure has been causally linked to a fall of eGFR following the nephron destruction induced by Cd [26]. In the same study, a 2-fold increase in the risk of hypertension was associated with the urinary Cd excretion of 1 µg/g creatinine and a blood Cd of 0.61 µg/L. In line with such findings, herein, we observed an inverse correlation between a SBP and C_cr_-normalized Cd excretion rate (E_Cd_/C_cr_) (*r* = −0.227) (Table 2). A rise in SBP as the eGFR falls explains a universally high prevalence of hypertension among those with a low eGFR. (eGFR ≤ 60 mL/min/1.73 m^2^). Interestingly, a rapid fall in eGFR (≥3 mL/min/1.73 m^2^ per year) has been causally linked to Cd excretion in a prospective cohort study from Switzerland [27], but an effect of the eGFR decline on blood pressure was not investigated.

Another notable result of the present work was that the risk of hypertension among participants was also influenced by hyperglycemia (FPG ≥ 110 and ≥126 mg/dL) (Table 5 and Table 6). In a bivariate analysis, the FPG correlated positively with SBP (*r* = 0.250), albumin excretion (*r* = 0.273), and Cd/Pb exposure (*r* = 0.181) (Table 2). The increased risk of hypertension among those with hyperglycemia may be a consequence of kidney damage, assessed with elevated levels of albumin excretion. A nonlinear relationship was observed between FBG and ACR in a representative U.S. population [28]. In a Chinese prospective cohort of non-diabetics, the risk of incident albuminuria rose 71% per every 18 mg/dL increment of FPG levels [29].

A higher risk of hypertension was associated with an elevation in ACR within the normal range in a meta-analysis, and ACR was suggested to be a predictor of incident hypertension in the general population [30]. Earlier studies on a Japanese population observed an increased risk of hypertension and a large decrease in the eGFR among those with an elevated E_β2M_/E_cr_, but environmental exposure to toxic metals was not investigated [31,32]. Increased blood pressure was associated with serum Cd in a recent cross-sectional study on the Japanese general population [33].

### 3.2. The SH3B-β_2_M Axis: A Novel Blood Pressure Regulator

The protein β_2_M is a non-polymorphic and non-glycosylated low-molecular-weight protein, forming an extracellular domain of the class I human leukocyte antigen or class I major histocompatibility complex, which is shed into the blood stream [34]. Its involvement in blood pressure control and hypertension development was deduced from a genome-wide association [23], single nucleotide polymorphism in the SH2B3 locus, which encodes for the regulator of cytokine signaling and cell proliferation, and a human longitudinal study [35]. Data from knockout mouse models have provided additional support to the involvement of SH3B-β_2_M axis in hypertension development plus kidney damage [35]. Comparing participants in top plasma β_2_M to the bottom quartile, the prevalence and incidence of hypertension among the participants in the Framingham Heart Study rose 29% and 59%, respectively [35].

In the present study, we found that serum β_2_M correlated with both FPG and SBP (Table 2). The SBP vs. [β_2_M]_s_ was particularly strong in those with a low eGFR (Figure 2), and the odds of having a high serum β_2_M fell 4% for every 1 mL/min/1.73 m^2^ higher eGFR in the regression analysis (Table 3). A high serum β_2_M was four times more prevalent in the diagnosed diabetics group (Table 3). FPG ≥ 110 and ≥126 mg/dL both were more prevalent in those with a high serum β_2_M (Table 4) and those with albuminuria (Table 5 and Table 6).

Subgroup analysis indicated that E_alb_/C_cr_ vs. SBP was tighter in the high serum β_2_M (R^2^ = 0.155) than the low serum β_2_M group (R^2^ = 0.099) (Figure 3a. This may reflect the independent effect of β_2_M (*SH2B3*-β_2_M axis) on SBP. It may also reflect more extensive kidney damage in those with [β_2_M]_s_ ≥ 5 mg/L. In agreement with our study, an investigation from China observed elevated serum β_2_M levels in patients with diabetes, together with a 17% increase in the prevalence of left ventricular hypertrophy per one standard deviation increase in serum β_2_M [36]. Furthermore, elevated serum β_2_M levels were associated with diabetes-related mortality and DKD [37,38]. Collectively, these findings suggest serum β_2_M as a potential biomarker for cardiovascular–kidney–metabolic (CKM) syndrome.

### 3.3. Mediating Effects of Cd on Serum β_2_M and SBP

We used the Baron and Kenny method to determine whether the effects of Cd on serum β_2_M and blood pressure (SBP/DBP) were direct/indirect or mediated by kidney tubular toxicity, assessed with the excretion of β_2_M (E_β2M_/C_cr_). We found that a statistically significant effect of Cd on serum β_2_M was mediated by E_β2M_/C_cr_ with a tendency for a direct effect; the *p*-value for its direct effect (c’) was 0.051 (Figure 2). The direct effect of Cd on the levels of circulating β_2_M requires confirmation with a larger sample group.

We also found that an effect of Cd on SBP, not DBP, was totally mediated by E_β2M_/C_cr_ (Figure 3). Thus, tubular dysfunction (E_β2M_/C_cr_) appeared to mediate the effects of Cd on both SBP and serum β_2_M concentrations.

To the best of our knowledge, the present study is the first to investigate a novel pathway of blood pressure control in Cd/Pb-exposed people with and without diabetes. Our findings are consistent with the role of serum β_2_M in the incidence and prevalence of hypertension recorded in the Framingham Heart Study and the functionality of the *SH2B3*-β_2_M axis for blood pressure homeostasis [23,35].

### 3.4. Strengths and Limitations

We used a cross-sectional design with a Thai cohort to explore how Cd and Pb induce kidney tubular cell damage and accelerate DKD, emphasizing β_2_M and hypertension. The use of multiple indicators of exposure and outcomes (Cd/Pb in blood, Cd in urine, eGFR, FPG, SBP/DBP, Eβ_2_M, and Ealb) is the major strength. The focus on women was considered as a strength because women are an at-risk group; an increased risk of hypertension was associated with blood Cd as little as 0.4 µg/L in white and Mexican–American women but not in black women or white, black, or Mexican–American men [39]. Thus, women form a Cd-sensitive group, suitable for mechanistic investigation with a modest sample size. The limitations include a one-time-only assessment of Cd/Pb exposure and its outcomes plus the limited sample size, non-representativeness, and inability to adequately adjust smoking effects.

## 4. Materials and Methods

### 4.1. Data Sourcing

Individuals with and without diabetes were selected from a pre-existing cohort of 88 diabetes and 88 non-diabetes controls, conducted from June to December 2021 (approval number WUEC-20-132-01, 28 May 2020). The selection criteria for both cases and controls, matching strategies, and findings have been reported previously [24]. The pre-existing cohort employed a purposive sampling technique to recruit 100 individuals with diagnosed diabetes and 100 potential non-diabetic controls from administrative records of a health promoting center in Pakpoon Municipality, Nakhon Si Thammarat Province, Thailand.

In brief, the inclusion criteria for cases were residents of the Pak Poon municipality, aged 40 years or older, who attended annual health checkups, and who were diagnosed with type 2 diabetes. For the control group, exclusion criteria were non-resident status, pregnancy and/or breastfeeding, and hospital records or a physician’s diagnosis of an advanced chronic disease, including heart disease, stroke, and cancer.

Participants were provided with the study objectives, study procedures, potential risks, and benefits, and they gave written informed consent prior to participation. Structured interview questionnaires were used to collect sociodemographic data, educational attainment, occupation, health status, family history of diabetes, use of dietary supplements, alcohol consumption, and smoking status. After individuals with missing data were excluded, a total of 137 individuals (72 with diabetes and 65 without diabetes) were analyzed in the present study.

### 4.2. Collection of Blood and Urine Samples

Subjects were requested to fast overnight, and collection of blood and urine samples was undertaken at the Pakpoon health center on the morning of the following day. Morning voided urine samples were collected in acid-washed polypropylene collection cups. Blood samples for the glucose assay were collected in tubes containing heparin as an anticoagulant and fluoride as an inhibitor of glycolysis. Blood samples for Cd and Pb analysis were collected in separate tubes containing ethylene diamine tetra-acetic acid (EDTA) as an anticoagulant.

Blood and urine samples were kept on ice and transported within one hour to the laboratory at Walailak University, where samples of plasma and serum were prepared. To prevent the degradation of β_2_M in acidic conditions, an alkaline (NaOH) solution was added to adjust the pH of urine aliquots to >6 before storage. Aliquots of urine, whole blood, serum, and plasma were stored at −80 °C for later analysis.

### 4.3. Quantification of Exposure to Cd, Pb, and Biomarkers of Kidney Effects

We used the human beta-2 microglobulin/β_2_M ELISA pair set (Sino Biological Inc., Wayne, PA, USA) to determine urine and serum concentration of β_2_M, with a lower limit of detection of 3.13 pg/mL. The plasma glucose assay was based on the oxidase–peroxidase method (Glu Colorimetric Assay Kit, Elabscience, Catalog No: E-BC-K234-M, Houston, TX, USA) [40]. Assays of creatinine in urine and plasma were based on Jaffe’s alkaline picrate method, as described previously [41]. The urinary albumin assay was based on the immunoturbidimetric method [42,43]. The coefficient of variation (CV) for all blood and urine assays was within acceptable clinical chemistry standards.

Urinary and whole blood Cd and Pb concentrations were determined with GBC System 5000 graphite furnace atomic absorption spectrometry (AAS) (GBC Scientific Equipment, Hampshire, IL, USA) [44]. Standards with As, Be, Cd, Cr (VI), Hg, Ni, Pb, Se, and Tl were used to calibrate the instrument (Merck KGaA, Darmstadt, Germany). Reference urine metal levels 1, 2, and 3 (Lyphocheck, Bio-Rad, Hercules, CA, USA) were used for quality control, analytical accuracy, and precision assurance. When urinary and blood concentrations of Cd and Pb were less than their detection limits, the concentration assigned was the detection limit value divided by the square root of 2 [45].

### 4.4. Assessment of Simultaneous Cd/Pb Exposure

To evaluate effects of simultaneous Cd and Pb exposures, subjects were grouped based on blood Cd and blood Pb levels. Accordingly, each subject was assigned to the Cd/Pb exposure category 1, 2, or 3 using her/his blood Cd and blood Pb levels. Respective Cd/Pb exposure categories 1, 2, and 3 were defined as blood Cd and blood Pb levels ≤ median, blood Cd or blood Pb levels ≥ median, and blood Cd plus blood Pb levels were above the median. The median for blood Cd was 0.3 µg/L and the median for blood Pb was 2.12 µg/dL. There were 44, 54, and 39 subjects with the Cd/Pb exposure categories 1, 2, and 3, respectively.

### 4.5. Calculation and Cut-Off Values for Albuminuria

Estimated GFR (eGFR) was computed with Chronic Kidney Disease Epidemiology Collaboration (CKD-EPI) equations [46]. CKD stages 1, 2, 3, 4, and 5 corresponded to eGFR of 90–119, 60–89, 30–59, 15–29, and <15 mL/min/1.73 m^2^, respectively.

In the present study, urine samples were collected at a single time point (voided urine). This necessitated a correction for interindividual differences in urine volume (dilution). To achieve this, we normalized excretion of Cd (E_Cd_) and albumin (E_alb_) to creatinine excretion (E_cr_) and creatinine clearance (C_cr_), using the below equations.

E_x_/E_cr_ = [x]_u_/[cr]_u_, where x = Cd or alb; [x]_u_ = urine concentration of x (mass/volume) and [cr]_u_ = urine creatinine concentration (mg/dL). E_x_/E_cr_ was expressed as an amount of x excreted per g of creatinine. Albumin-to-creatinine ratio (ACR) is a well-known expression of E_alb_/E_cr_.

E_x_/C_cr_ = [Cd]_u_[cr]_p_/[cr]^u^, where x = Cd or alb; [x]_u_ = urine concentration of x (mass/volume); [cr]_p_ = plasma creatinine concentration (mg/dL); and [cr]_u_ = urine creatinine concentration (mg/dL). E_x_/C_cr_ was expressed as an amount of x excreted per volume of the glomerular filtrate [47].

For E_alb_/E_cr_ (ACR) data, albuminuria is defined as ACR value ≥ 20 and 30 mg/g creatinine in men and women, respectively, [1,2,3]. A higher cut-off value for ACR in women is to compensate for their universally lower E_cr_ values due to having a smaller muscle mass than men. In comparison, C_cr_-normalization is not affected by muscle mass. Hence, for E_alb_/C_cr_ data, albuminuria is defined as E_alb_/C_cr_ values ≥ 0.2 mg/L filtrate in both men and women.

### 4.6. The Causal Inference Analysis

The Baron and Kenny method [48,49,50] was employed to explore the causal connection between outcomes of Cd-induced kidney tubular dysfunction and blood pressure control by the *SH2B3* pathway. The mediation models are depicted below (Figure 1).

In theory, the variability in serum β_2_M levels depends on the rate of its production by nucleated cells in the body and its degradation within the proximal tubular (PT) cells of the kidneys [34]. An increase in the excretion of β_2_M (E_β2M_/C_cr_) is a well-known consequence Cd toxicity in the PT cells.

Model A tests for the potential direct effect of E_Cd_/C_cr_ on β_2_M production and the indirect effect of E_Cd_/C_cr_ on β_2_M degradation by the kidneys. Model B tests for the potential direct and indirect effects of E_Cd_/C_cr_ on SBP/DBP. Given a significant correlation between E_Cd_/C_cr_ and Cd/Pb exposure levels (Table 2), the observed effects reported in Figure 2 and Figure 3 may have reflected a combined effect of Cd/Pb exposure. Due to the low Pb exposure levels among participants, their urinary Pb excretions were below detection limits. Excretion of Cd is an indicator of cumulative kidney burden. Bone Pb is a reliable marker to assess long-term Pb exposure.

### 4.7. Statistical Analysis

Data were analyzed with IBM SPSS Statistics 21 (IBM Inc., New York, NY, USA). The variation in any continuous variable and differences in percentages across the tertiles of [β_2_M]_s_ (Table 1) were assessed by the Kruskal–Wallis’s test and the Pearson chi-squared test, respectively. Spearman’s rank correlation analysis was employed to produce the correlation matrices of ten variables: [β_2_M]_s_, age, BMI, [Glc]_p_, SBP, DBP, eGFR, E_alb_/C_cr_, E_β2M_/C_cr_, E_Cd_/C_cr_, and Cd/Pb exposure category (Table 2). The one-sample Kolmogorov–Smirnov test was used to assess deviation from a normal distribution of any continuous variable. Logarithmic transformation was applied to fasting plasma glucose concentration ([Glc]_p_) and the excretion rate of albumin and Cd (E_alb_/E_cr_ and E_Cd_/C_cr_) that showed rightward skewing before they were subjected to parametric statistics analyses, scatterplots, and linear regressions (Figure 1, Figure 2 and Figure 3).

The prevalence odds ratio (POR) values for [β_2_M]_s_ ≥ 5 mg/L, FPG ≥ 110 and ≥126 mg/dL, hypertension, and albuminuria were determined by multivariable logistic regression modeling with adjustment for covariates (Table 3, Table 4, Table 5 and Table 6). The common variables incorporated in models as covariates were age, BMI, and smoking, given that they all have an impact on kidney function. In addition, smoking could also be a source of Cd and Pb exposure.

## 5. Conclusions

Hyperglycemia, a falling eGFR, and rising SBP could be the toxic manifestation of chronic exposure to low-level Cd and Pb, leading to hypertension, kidney damage, and albuminuria. For the first time, we present evidence that causally links circulating β_2_M to rising SBP and kidney damage, associated with Cd/Pb exposure. These findings underscore the coexistence of metabolic and kidney disease, recognized by the American Heart Association as cardiovascular–kidney–metabolic (CKM) syndrome. Our work bridges Cd/Pb exposure, β_2_M, and the SH3B pathway—a novel framework for diabetic kidney disease pathogenesis. It expands on genomic studies and emphasizes β_2_M as a biomarker for CKM syndrome.

## Data Availability

All data are contained within this article.

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
