# Peer review of "Enhanced Kidney Damage in Individuals with Diabetes Who Are Chronically Exposed to Cadmium and Lead: The Emergent Role for β_2_-Microglobulin"

_ijms, 2025, doi:10.3390/ijms26189208_

Round 1
Reviewer 1 Report
Comments and Suggestions for Authors
The authors present a study on kidney damage in individuals with diabetes exposed to cadmium and lead, with a potential role for beta microglobulin. The subject is interesting and potentially clinically important for individuals who already have diabetes and live in areas with heightened exposure to heavy metals. However, the presentation of the study could be strengthened by 1) a clear presentation of a hypothesis and 2) presentation of statistical analyses and resulting findings that directly address the hypothesized relationships. For example, from the title of the study, it is understood that lead and cadmium are the exposure (independent) variables and that kidney damage is the outcome (independent) variable, whereas beta microglobulin is the potential mechanism or mediating variable. Yet, the analyses are not presented in a way that would test these implied relationships. One way to do this would be to employ mediation analysis. There is also quite a large focus on hypertension, both in the introduction and in the presentation of results, without clarifying on why this is done. The detailed comments that follow are recommendations for addressing some of the limitations outlined above.
Abstract:
Line 19 (and throughout the manuscript): Instead of using the term "diabetics", consider this wording "72 individuals with diabetes and 65 controls without diabetes".
Lines 25-6: It is unclear what categories 2 and 3 of Cd/Pb exposure are. Could you define these first? Consider revising the background and objective to be more succinct to fit this additional information.
Introduction:
Line 37: it might be better to say "where" than "in whom"
The authors are encouraged to expand the introduction to clearly define the scope of the problem. For example, why is it a concern that little is known about the effects of Cd and Pb in individuals with diabetes? What heightened risk, if any, may be expected in those individuals?
Also, note that the title of the paper implies that kidney damage will be the endpoint and cadmium/lead exposure are the independent variables, with beta microglobulin acting as the mechanism. Some explanation for why this is the hypothesized relationship needs to be included.
Second, if hypertension and diabetes are both causes of kidney failure, why is the study investigating risk of hypertension in individuals with diabetes? Does diabetes come first and then hypertension in disease progression? Some more information on disease development would be very helpful both to set up the problem under study and to aid in the interpretation of findings.
Lines 55-60: What is the hypothesis of this study? What is the exposure and what is the outcome? Please clearly define/identify these.
Results:
Section 2.1 and Table 1: This is a matched case-control study design. Please provide a participant characteristics table that clearly compares the characteristics of cases and controls.
Line 74: Here the study is defined as "cohort" but elsewhere, it is considered a case-control study. Please be clear and consistent about the study design.
Table 2: Instead of or in addition to the footnote, please provide information on ECd/Pb categories in the methods section for easier location by interested readers. Also, it is somewhat unusual to use a spearman correlation between a 3-level categorical variable and a continuous variable. It would make more sense to present all continuous variables by the 3 categories of the combined exposure variable, and use one-way ANOVA to test differences overall and Scheffe statistic for between-category differences. This is particularly important if Cd/Pb are the exposures of interest in this study.
Another consideration with this analysis is that all cases and controls are grouped together. It should really be conducted separately. First, because the matched cases and controls are not independent of each other, given the selected study design. Second, because grouping them may obscure important differences based on diabetes disease status.
Figure 1: This comment goes all the way back to the introduction and the presentation of objectives. Because it is not clear which way the authors hypothesize the relationships to be going, it is difficult to understand why beta microglobulin is presented on the x axis and low vs. normal eGFR are presented separately. The presentation of data should follow a logical pathway that shows the relationship between exposure(s) and the outcome(s). The case-control design of this study is also lost in the presentation of these findings yet it is important, particularly because (per paper title) the authors want to see what these relationships look like for people with diabetes.
The comment on Figure 2 is similar to above. Why is blood pressure on the X axis when it is not the exposure of interest in this study?
Table 3: This analysis uses a matched-case control design to investigate relationships for which the case-control study was not designed. While the cases and controls are combined here, they are not really independent. To stay true to the original design (and the implied study question), analyses should be performed separately (stratified models) by diabetes diagnosis status.
Please provide the N for this analysis. In the methods section, please explain how the covariates were selected. For example, why is E-Cd/Cr included here but not E-Pb? or E-Cd/Pb? Again, having a clear hypothesis would help guide all analyses in this study.
Table 4: Similar comments as for Table 3. Why is beta microglobulin the exposure/independent variable here? Why is Cd/Pb category included in this but not other models?
Discussion:
There is no inclusion of strengths and limitations.
This section may need to be revised once hypothesized relationships and modeling strategy is clarified.
Materials and Methods:
Section 4.1: Could the authors provide more information on recruitment. How was it conducted, by whom, and what information was initially provided to the individuals being recruited.
For the individuals with diabetes, was the fasting plasma glucose mentioned on lines 263-4 available at the time of recruitment? or does this refer to a history of having an elevated Glc?
Also, while the definition of diabetes is provided, the definition of the control group is less clear. Lack of advanced chronic disease does not exclude the presence of diabetes or pre-diabetes. Was a fasting plasma glucose level available for these individuals at the time of recruitment? Does everyone have this tested at the time of the annual checkup?
Furthermore, please provide more information on the population being served by the health promoting center. Is healthcare provided free of charge?
Other information that is needed for better understanding of the study includes:
1. How many people were invited to participate and how many declined (what was the participation rate).
2. How was the sample size for this study calculated?
Lines 262-3: How was matching conducted? Was it at the time of the recruitment? Or was a "pool" of potential controls available earlier? To match on 3 separate criteria, a large number of controls must have been either screened or available. This is not clearly described.
How was matching on age and residential locality conducted? Was age matched 1-1 or within age categories?
Section 4.2: Could the authors provide more information on the make of the blood tubes?
Also, urine samples are mentioned in this section but there is nothing to indicate how urine was collected ann for what purpose. Please supply this information.
Finally, both serum and plasma are mentioned. How were they obtained? Only a fluoride coated tube is mentioned. How would that yield both plasma and serum? Why are both needed? What was measured in them?
It would be helpful to provide an overview sentence at the beginning of this section to say what samples were obtained from the participants and for what purpose.
Section 4.3: Please provide % CVs for all these assays.
Lines 331-3: The information on multivariable modeling would benefit from providing a list of covariates and explaining how/why they were selected and are considered appropriate for the study question/hypothesis.
Hypertension or blood pressure may need to be a model covariate.
Other missing information that is important for the interpretation of this study is whether all participants completed all study procedures, whether there was any missing data, and how the authors dealt with this statistically. For example, from the abstract it looks like there were 72 cases and 65 controls. It looks like perfect 1:1 matching was not possible in this study. Did some controls get matched to more than 1 case?
Author Response
Reviewer 1
Comments and Suggestions
The authors present a study on kidney damage in individuals with diabetes exposed to cadmium and lead, with a potential role for beta microglobulin. The subject is interesting and potentially clinically important for individuals who already have diabetes and live in areas with heightened exposure to heavy metals. However, the presentation of the study could be strengthened by 1) a clear presentation of a hypothesis and 2) presentation of statistical analyses and resulting findings that directly address the hypothesized relationships. For example, from the title of the study, it is understood that lead and cadmium are the exposure (independent) variables and that kidney damage is the outcome (independent) variable, whereas beta microglobulin is the potential mechanism or mediating variable. Yet, the analyses are not presented in a way that would test these implied relationships. One way to do this would be to employ mediation analysis. There is also quite a large focus on hypertension, both in the introduction and in the presentation of results, without clarifying on why this is done. The detailed comments that follow are recommendations for addressing some of the limitations outlined above.
RESPONSES: We thank the reviewer for insightful comments and guidance for improvement our manuscript. Accordingly, our paper has been revised extensively, Major undertakings are listed below
- Hypothesis has now been explicitly stated in the abstract (lines 15-18).
- The Baron and Kenny mediation analysis has been applied to infer serum β2M and the excretion of β2M (an indicator of tubular toxicity indicator) as mediators of Cd effects on SBP.
- Results of the mediation analysis are presented (lines 116-130). The method is described under subsection, 4.5. The causal inference analysis (lines 389-397) and results are discussed (lines 280-294)
- The introduction has been rewritten to better reflect study focuses, objectives, and the working hypothesis is stared (lines 58-60).
Abstract:
Line 19 (and throughout the manuscript): Instead of using the term "diabetics", consider this wording "72 individuals with diabetes and 65 controls without diabetes".
Response: The suggestion has been undertaken.
Lines 25-6: It is unclear what categories 2 and 3 of Cd/Pb exposure are. Could you define these first? Consider revising the background and objective to be more succinct to fit this additional information.
Response: The category of Cd/Pb exposure levels have now been detailed in Section 4.4. Assessment of simultaneous Cd/Pb exposure (lines 358-366).
Introduction:
Point 1.1: Line 37: it might be better to say "where" than "in whom"
Response: The suggestion has been undertaken.
Point 1.2: The authors are encouraged to expand the introduction to clearly define the scope of the problem. For example, why is it a concern that little is known about the effects of Cd and Pb in individuals with diabetes? What heightened risk, if any, may be expected in those individuals?
Point 1.3: Also, note that the title of the paper implies that kidney damage will be the endpoint and cadmium/lead exposure are the independent variables, with beta microglobulin acting as the mechanism. Some explanation for why this is the hypothesized relationship needs to be included.
Point 1:4: Second, if hypertension and diabetes are both causes of kidney failure, why is the study investigating risk of hypertension in individuals with diabetes? Does diabetes come first and then hypertension in disease progression? Some more information on disease development would be very helpful both to set up the problem under study and to aid in the interpretation of findings.
Point 1.6: Lines 55-60: What is the hypothesis of this study? What is the exposure and what is the outcome? Please clearly define/identify these.
Responses to points 1.2-1.6
The introduction has been written to emphasis three key issues.
- Why the present study is focused on hypertension, diabetes, and women in Cd-related CKD research.
- Excretion of β2M as an indicator of tubular dysfunction and its new proposed role in mediating an effect of Cd on blood pressure.
- Working hypothesis.
Results:
Point 2.1: Section 2.1 and Table 1: This is a matched case-control study design. Please provide a participant characteristics table that clearly compares the characteristics of cases and controls.
Response: The description as a matched case-control study design was in error. Where applicable, we have now been corrected as “cohort”.
Pont 2.2: Line 74: Here the study is defined as "cohort" but elsewhere, it is considered a case-control study. Please be clear and consistent about the study design.
Response: Where applicable, a case-control study as now been changes to “cohort”.
Point 2:3 Table 2: Instead of or in addition to the footnote, please provide information on ECd/Pb categories in the methods section for easier location by interested readers. Also, it is somewhat unusual to use a spearman correlation between a 3-level categorical variable and a continuous variable. It would make more sense to present all continuous variables by the 3 categories of the combined exposure variable, and use one-way ANOVA to test differences overall and Scheffe statistic for between-category differences. This is particularly important if Cd/Pb are the exposures of interest in this study.
Response: Information on ECd/Pb exposure categories is now provided in Section 4.4. Assessment of simultaneous Cd/Pb exposure (lines 358-366). Excretion of β2M (Eβ2M) has been added to the correlation Table 2. The correlation of categorical variables (Cd/Pb exposure) are retained as it is coded in the order of relatively low, moderate, and high exposure levels.
Point 2.4: Another consideration with this analysis is that all cases and controls are grouped together. It should really be conducted separately. First, because the matched cases and controls are not independent of each other, given the selected study design. Second, because grouping them may obscure important differences based on diabetes disease status.
Response: We agree with the reviewer that cases and controls should be analyzed separately to address Cd/Pb exposure effects. We have published the results of case-control analysis [41]. In the present study, we analyzed diabetes and non-diabetes together as one group for mechanism study (to explore the potential role of β2M). For this reason, we grouped participants according to tertile of serum β2M (Table 1).
[41] Yimthiang, S, Pouyfung P, Khamphaya T, Kuraeiad S, Wongrith P, Vesey DA, Gobe GC, Satarug S. Effects of Environmental Exposure to Cadmium and Lead on the Risks of Diabetes and Kidney Dysfunction. Int J Environ Res Public Health. 2022 Feb 16;19(4):2259.
Point 2.4: Figure 1: This comment goes all the way back to the introduction and the presentation of objectives. Because it is not clear which way the authors hypothesize the relationships to be going, it is difficult to understand why beta microglobulin is presented on the x axis and low vs. normal eGFR are presented separately. The presentation of data should follow a logical pathway that shows the relationship between exposure(s) and the outcome(s). The case-control design of this study is also lost in the presentation of these findings yet it is important, particularly because (per paper title) the authors want to see what these relationships look like for people with diabetes.
Point 2.5: The comment on Figure 2 is similar to above. Why is blood pressure on the X axis when it is not the exposure of interest in this study?
Response to points 2.4 and point 2.5.
- New Figure 2 and new Figure 3 reporting mediation analysis have been inserted (lines 116-130) to explain why serum B2M is presented as a predictor of SBP (Figure 1).
- As for comments on old Figure 2, albuminuria is a marker of kidney damage, which may be a consequence of an elevation of blood pressure, induced by Cd/Pb. We did not investigate effect of Cd/Pb on albumin excretion in the present study. We, however, examined a mediating effect of blood pressure.
Point 2.6: Table 3: This analysis uses a matched-case control design to investigate relationships for which the case-control study was not designed. While the cases and controls are combined here, they are not really independent. To stay true to the original design (and the implied study question), analyses should be performed separately (stratified models) by diabetes diagnosis status.
Response: Please see response to point 2.4 above.
Point 2.7: Please provide the N for this analysis. In the methods section, please explain how the covariates were selected. For example, why is E-Cd/Cr included here but not E-Pb? or E-Cd/Pb? Again, having a clear hypothesis would help guide all analyses in this study.
Response: We selected ECd/Ccr because this parameter is an indicator of kidney burden of Cd or long-term exposure. This parameter correlates also with Cd/Pb exposure categories. Number of subjects has now been indicated in data tables, figures.
Point 2:8: Table 4: Similar comments as for Table 3. Why is beta microglobulin the exposure/independent variable here? Why is Cd/Pb category included in this but not other models?
Response: Excretion of β2M has been used widely to indicate tubular toxicity of Cd which result in reduced reabsorption of the protein and hence an increase in its excretion.
Discussion:
There is no inclusion of strengths and limitations. This section may need to be revised once hypothesized relationships and modeling strategy is clarified.
RESPONSE: Strengths and limitations have been provided (lines 295-306). The discussion section been expanded and revised extensively. Six additional supporting references are included.
Materials and Methods:
Point 3.1: Section 4.1: Could the authors provide more information on recruitment. How was it conducted, by whom, and what information was initially provided to the individuals being recruited.
Point 3.2: For the individuals with diabetes, was the fasting plasma glucose mentioned on lines 263-4 available at the time of recruitment? or does this refer to a history of having an elevated Glc?
Point 3:3: Also, while the definition of diabetes is provided, the definition of the control group is less clear. Lack of advanced chronic disease does not exclude the presence of diabetes or pre-diabetes. Was a fasting plasma glucose level available for these individuals at the time of recruitment? Does everyone have this tested at the time of the annual checkup?
Point 3.4: Furthermore, please provide more information on the population being served by the health promoting center. Is healthcare provided free of charge?
Responses to points 3.1-3.4
- Old Section 4.1 has been replaced by new section 4.1. Data sourcing (lines 308-327).
- To confirm DM diagnosis and DM free status of the controls, we measured FPG along with all other biomarkers reported in Table 1. We did not use FPG data recorded by the local health center.
- Thailand practices the universal health care nationwide with a minimal out of pocket fee
Point 3.5. Other information that is needed for better understanding of the study includes:
- How many people were invited to participate and how many declined (what was the participation rate).
- How was the sample size for this study calculated?
Point 3.6: Lines 262-3: How was matching conducted? Was it at the time of the recruitment? Or was a "pool" of potential controls available earlier? To match on 3 separate criteria, a large number of controls must have been either screened or available. This is not clearly described.
Point 3.7 How was matching on age and residential locality conducted? Was age matched 1-1 or within age categories?
Response to points 3.5-3.7
- Please kindly see a revised Section 4.1 (lines 308-327).
Point 3.8: Section 4.2: Could the authors provide more information on the make of the blood tubes?
Point 3.9. Also, urine samples are mentioned in this section but there is nothing to indicate how urine was collected and for what purpose. Please supply this information.
Point 3.10. Finally, both serum and plasma are mentioned. How were they obtained? Only a fluoride coated tube is mentioned. How would that yield both plasma and serum? Why are both needed? What was measured in them?
Point 3.11. It would be helpful to provide an overview sentence at the beginning of this section to say what samples were obtained from the participants and for what purpose.
Point 3.12. Section 4.3: Please provide % CVs for all these assays.
Response to points 3.8-3.12.
- Details on blood/urine sample collection and their analysis have been addressed in the revised Sections 4.2 and 4.3.
Pint 3.13. Lines 331-3: The information on multivariable modeling would benefit from providing a list of covariates and explaining how/why they were selected and are considered appropriate for the study question/hypothesis.
Point 3.14. Hypertension or blood pressure may need to be a model covariate.
Response to points 3.13 and 3.14.
- Covariates have been reported in multivariable modeling results (Tables 3-6).
- We hypothesize that rising blood pressure is one of the effects of Cd/Pb exposure and we tested our hypothesis by mediation analysis according to the reviewer suggestion.
- Please see Section 3.3. Mediating effects of Cd on serum β2M and SBP in the Discussion (lines 280-294).
Point: 3.15 Other missing information that is important for the interpretation of this study is whether all participants completed all study procedures, whether there was any missing data, and how the authors dealt with this statistically. For example, from the abstract it looks like there were 72 cases and 65 controls. It looks like perfect 1:1 matching was not possible in this study. Did some controls get matched to more than 1 case?
- Response: Kindly see Section 4.1. Data sourcing (lines 308-327), where issues raised have been addressed.
Reviewer 2 Report
Comments and Suggestions for Authors
Dear author:
This study investigates the mechanisms of kidney damage in diabetic patients chronically exposed to low-dose cadmium (Cd) and lead (Pb), emphasizing the role of serum β2-microglobulin (β2M). Using a cross-sectional design with a Thai cohort (72 diabetics and 65 non-diabetic controls), the authors explore how environmental pollutants (Cd/Pb) induce renal tubular toxicity and accelerate diabetic kidney disease (DKD)
This manuscrpt bridges Cd/Pb exposure, β2M, and the SH3B pathway—a novel framework for DKD pathogenesis. This expands on prior genomic studies (e.g., Huan et al. 2015; Keefe et al. 2019) and underscores β2M as a biomarker for cardiorenal-metabolic (CKM) syndrome.
Specific comments:
Limited Sample Size and Representativeness: The cohort (N=137) is small and predominantly female (78.1%, Table 1), reducing generalizability to males or broader populations.
Cross-Sectional Design: Precludes causal conclusions (e.g., β2M elevation may result fromrather than causekidney damage). Single-time urine samples (Section 4.4) weaken biomarker reliability.
Inadequate Adjustment for Confounders: Smoking status (a known modifier of Cd/Pb toxicity) is underadjusted in regression models (Tables 5–6), potentially inflating risk estimates (e.g., non-smoker hypertension POR = 7.9).
Author Response
Reviewer 2
Comments and Suggestions
This study investigates the mechanisms of kidney damage in diabetic patients chronically exposed to low-dose cadmium (Cd) and lead (Pb), emphasizing the role of serum β2-microglobulin (β2M). Using a cross-sectional design with a Thai cohort (72 diabetics and 65 non-diabetic controls), the authors explore how environmental pollutants (Cd/Pb) induce renal tubular toxicity and accelerate diabetic kidney disease (DKD)
This manuscript bridges Cd/Pb exposure, β2M, and the SH3B pathway—a novel framework for DKD pathogenesis. This expands on prior genomic studies (e.g., Huan et al. 2015; Keefe et al. 2019) and underscores β2M as a biomarker for cardiorenal-metabolic (CKM) syndrome.
RESPONSE: We thank the reviewer for evaluating our work and comments for further improvement of a manuscript.
Specific comments:
Comment 1: Limited Sample Size and Representativeness: The cohort (N=137) is small and predominantly female (78.1%, Table 1), reducing generalizability to males or broader populations.
Comment 2: Cross-Sectional Design: Precludes causal conclusions (e.g., β2M elevation may result from rather than cause kidney damage). Single-time urine samples (Section 4.4) weaken biomarker reliability.
Responses to comments 1 and 2: We acknowledge the limitations due to sample size, the study design and one-time assessment of exposure and effects. We have now added the mediation analysis results in new Figures 2 and 3 (lines 116-130). Thus, evidence has been presented to causally link Cd exposure to kidney tubular toxicity, serum β2M and rising SBP.
Comment 3: Inadequate Adjustment for Confounders: Smoking status (a known modifier of Cd/Pb toxicity) is under adjusted in regression models (Tables 5–6), potentially inflating risk estimates (e.g., non-smoker hypertension POR = 7.9).
RESPONSE: We agree with the reviewer’s notion that smoking was not adequately adjusted due mostly to a small number of smokers and a lack of quantitative smoking measures like urinary cotinine data. We have listed this as one of the limitations. We however wish to draw the reviewer’s attention that our results were in line with those found in a large study on the U.S. general population (lines 50-53), quoted below.
“A cross-sectional study on U.S. population observed that Pb exposure may have increased risk of CKD, especially in women with body mass index (BMI) higher than 25 kg/m2 and diabetes, who were non-smokers [20].”
[20] Zhao, H.; Yin, R.; Wang, Y.; Wang, Z.; Zhang, L.; Xu, Y.; Wang, D.; Wu, J.; Wei, L.; Yang, L.; Zhao, D. Association between blood heavy metals and diabetic kidney disease among type 2 diabetic patients: a cross-sectional study. Sci. Reports 2024, 14, 26823.
Comment 4: The English could be improved to more clearly express the research.
RESPONSE: We have carefully read through a paper for errors, and undertaken necessary rewordings for clarity.
Reviewer 3 Report
Comments and Suggestions for Authors
This study aims to investigate the kidney impairment in diabetic patients exposed to low concentrations of Cd and Pb in addition to its link with β2-microglobulin (β2M). This study is interesting; however, some concerns should be considered to enhance the quality of the manuscript as follows:
- Abstract: The aim of this study should be clearly described. I presume the authors should try to simplify the abstract since it includes many quantitative data with p value.
- Intro: line 44: by Verzelloni et al. [15], please do not mention the year. The authors should discuss various deleterious effects caused by exposure to Cd and Pb and the common mechanism whether direct or indirect mechanism derived from previous literatures. The authors are encouraged to add a figure showing the flow procedures applied in this study.
- Results: Line 74: As shown in Table 1, …… (please correct). I have no further comments.
- Discussion: It is well written, but the authors should compare their findings with previous studies in light of available data. Limitations of this study should be discussed in detail.
- Materials and Methods: Line 285: Please mention the Cat. No. of the kit.
- Conclusion: It is too short; please elaborate on this section and highlight the key findings.
- The plagiarism of the manuscript is 34% based on MDPI analysis; please address this issue.
Author Response
Reviewer 3
Comments and Suggestions
This study aims to investigate the kidney impairment in diabetic patients exposed to low concentrations of Cd and Pb in addition to its link with β2-microglobulin (β2M). This study is interesting; however, some concerns should be considered to enhance the quality of the manuscript as follows:
RESPONSE: We thank the reviewer for evaluating our work and comments for further improvement of a manuscript.
Point 1: Abstract: The aim of this study should be clearly described. I presume the authors should try to simplify the abstract since it includes many quantitative data with p value.
Response: The purpose of the present study has been defined in the abstract (lines 15-17), quote below. All p-values have been deleted.
“The present study tests the hypothesis that the environmental pollutants, cadmium (Cd) and lead (Pb), by increasing plasma β2M level, promote the development of hypertension and progression of diabetic kidney disease.”
Point 2: Intro: line 44: by Verzelloni et al. [15], please do not mention the year. The authors should discuss various deleterious effects caused by exposure to Cd and Pb and the common mechanism whether direct or indirect mechanism derived from previous literatures. The authors are encouraged to add a figure showing the flow procedures applied in this study.
Response: The year cited has been deleted. In the present work, we have addressed Cd and Pb toxicity particularly in the kidneys of people with diabetes. To-date, there have been only a few studies conducted on this subgroup of population (ref. 19-21) and findings from these studies have been summarized (lines 49-55). The key message was that kidney damage in patients with diabetes was more severe than those without diabetes even though they were similarly exposed to Cd/Pb.
[19] Oosterwijk, M.M.; Hagedoorn, I.J.M.; Maatman, R.G.H.J.; Bakker, S.J.L.; Navis, G.; Laverman, G.D. Cadmium, active smoking and renal function deterioration in patients with type 2 diabetes. Nephrol. Dial. Transplant. 2023, 38, 876–883.
[20] Zhao, H.; Yin, R.; Wang, Y.; Wang, Z.; Zhang, L.; Xu, Y.; Wang, D.; Wu, J.; Wei, L.; Yang, L.; Zhao, D. Association between blood heavy metals and diabetic kidney disease among type 2 diabetic patients: a cross-sectional study. Sci. Reports 2024, 14, 26823.
[21] Barregard, L.; Bergström, G.; Fagerberg, B. Cadmium, type 2 diabetes, and kidney damage in a cohort of middle-aged women. Environ. Res. 2014, 135, 311-316.
Point 3: Results: Line 74: As shown in Table 1, …… (please correct). I have no further comments.
Response: The correction has been undertaken.
Point 4: Discussion: It is well written, but the authors should compare their findings with previous studies in light of available data. Limitations of this study should be discussed in detail.
Response: Findings from previous studies have been included (lines 232-234, 247-251). The discussion section has been expanded and revised extensively. Six additional supporting references are included. The limitations of our study have been declared (lines296-306).
Point 5: Materials and Methods: Line 285: Please mention the Cat. No. of the kit.
Response: The catalog no of the has been provided (line 345).
Point 6: Conclusion: It is too short; please elaborate on this section and highlight the key findings.
Response: Conclusion has been expanded to include findings from additional data analysis using the Baron and Kenny method.
Point 7: The plagiarism of the manuscript is 34% based on MDPI analysis; please address this issue.
Response: We will reduce the plagiarism to an acceptable level.
Round 2
Reviewer 1 Report
Comments and Suggestions for Authors
The authors have addressed many of my comments, but other points and concerns remain to be clarified. See below for details
Abstract
Line 29: Could the authors find all instances where the term “diabetics” is used and change it to “individuals with diabetes”/
Introduction
Lines 49-56: Since the authors mention in their responses to reviewers that they have previously analyzed the link between environmental exposure and kidney disfunction in the case-control study, cold that study be included in the introduction? It would also be helpful to explain what new information the current analysis brings in. Is it to investigate the mechanisms?
[41] Yimthiang, S, Pouyfung P, Khamphaya T, Kuraeiad S, Wongrith P, Vesey DA, Gobe GC, Satarug S. Effects of Environmental Exposure to Cadmium and Lead on the Risks of Diabetes and Kidney Dysfunction. Int J Environ Res Public Health. 2022 Feb 16;19(4):2259.
Methods:
Diagram 1 seems to describe the mediation by E-beta macroglobulin between ECd and serum beta macroglobulin (Model A) and between ECd and blood pressure (Model B). It is unclear why Pb exposure or diabetes is not accounted for in this relationship given that the hypothesis is that Cd/Pb and/or diabetes will increase plasma beta-macroglobulin.
Per lines 59-61 in the introduction:
We hypothesize that Cd/Pb exposure and/or diabetes (hyperglycemia) increase plasma β2M levels, which, in turn, raises blood pressure and induces kidney tubular cell damage.
It also remains unclear how results in table 4, which test the association of serum beta-macroglobulin with odds of hyperglycemia. Given the stated hypothesis, hyperglycemia should be the exposure and beta-macroglobulin the outcome. This model seems to imply the reverse.
Could the authors include brief statements in sections 2.4 and 2.5 to explain how the results in Tables 4-6 address the hypothesis? This additional explanation would help readers not closely familiar with the plasma/urinary indicators to follow the numerous statistical tests that are being conducted.
Could the authors clarify why the mediation models utilize Cd only but logistic models utilize Cd/Pb co-exposure?
The information on model covariates in the models/tables themselves is helpful but please include this information also in the methods section, and explain how those variables were selected.
Author Response
Reviewer 1
Comments and Suggestions
The authors have addressed many of my comments, but other points and concerns remain to be clarified. See below for details
RESPONSE: We thank the reviewer for giving us feedback and further guidance to improve our manuscript. Changes to the text are indicated by red lettering. We hope that we have clarified the methodological and theoretical issues that the reviewer has raised.
Abstract
Line 29: Could the authors find all instances where the term “diabetics” is used and change it to “individuals with diabetes”/
Response: We have looked for the word “diabetics” in an entire paper and replaced it with “individuals with diabetes”. The title of a paper has been changed to read, Enhanced Kidney Damage in Individuals with Diabetes Who Are Chronically Exposed to Cadmium and Lead: The Emergent Role for β2-Microglobulin.
Introduction
Lines 49-56: Since the authors mention in their responses to reviewers that they have previously analyzed the link between environmental exposure and kidney disfunction in the case-control study, could that study be included in the introduction? It would also be helpful to explain what new information the current analysis brings in. Is it to investigate the mechanisms?
[41] Yimthiang, S, Pouyfung P, Khamphaya T, Kuraeiad S, Wongrith P, Vesey DA, Gobe GC, Satarug S. Effects of Environmental Exposure to Cadmium and Lead on the Risks of Diabetes and Kidney Dysfunction. Int J Environ Res Public Health. 2022 Feb 16;19(4):2259.
Response: The reviewer’s suggestions have been incorporated in the text (lines 60-66). The former reference 41, now reference 25, has been cited here.
Methods:
Point 1. Diagram 1 seems to describe the mediation by E-beta microglobulin between ECd and serum beta microglobulin (Model A) and between ECd and blood pressure (Model B). It is unclear why Pb exposure or diabetes is not accounted for in this relationship given that the hypothesis is that Cd/Pb and/or diabetes will increase plasma beta-microglobulin.
Per lines 59-61 in the introduction:
We hypothesize that Cd/Pb exposure and/or diabetes (hyperglycemia) increase plasma β2M levels, which, in turn, raises blood pressure and induces kidney tubular cell damage.
Response: We have constructed two new paragraphs to fully explain Models A and B (lines 406-417). We have indicated that the results of mediation analyses, provided in Figures 2 and 3, could be interpreted as a combined effect of Cd/Pb exposure.
Point 2. It also remains unclear how results in table 4, which test the association of serum beta-microglobulin with odds of hyperglycemia. Given the stated hypothesis, hyperglycemia should be the exposure and beta-microglobulin the outcome. This model seems to imply the reverse.
Point 3. Could the authors include brief statements in sections 2.4 and 2.5 to explain how the results in Tables 4-6 address the hypothesis? This additional explanation would help readers not closely familiar with the plasma/urinary indicators to follow the numerous statistical tests that are being conducted.
Response to Points 2 and 3:
We have changed to wordings in sections 2.4 and 2.5 to reflect below explanations.
- In theory, the serum β2M concentration vary with its production by nucleated cells and its degradation by the kidney tubular cells (lines 406-409). Table 3 reports the variables that affected serum β2M levels, which included eGFR (a clinical measure of kidney function) and the diagnosis of diabetes (lines 152-162).
- We conducted the logistic regression models for FPG ≥110 and FPG ≥126 mg/dL (Table 4, lines 167-177) to ascertain that an increased odds of the serum β2M ≥ 5 mg/L in the participants, who were diagnosed with diabetes (Table 3), could indeed be attributable to an elevation in fasting plasma glucose (FPG ≥110 and FPG ≥126 mg/dL).
- Tables 5 and 6 report results of logistic regression models showing associations of hyperglycemia with hypertension and albuminuria which were independent of effects of Cd/Pb exposure. These results have been discussed with supporting literatures (lines 243-258).
Point 4. Could the authors clarify why the mediation models utilize Cd only but logistic models utilize Cd/Pb co-exposure?
Response: Please see responses to Point 1 above.
Point 5: The information on model covariates in the models/tables themselves is helpful but please include this information also in the methods section, and explain how those variables were selected.
Response: The reviewer’s suggestions have been undertaken (lines 432- 435).

Reviewer 2 Report
Comments and Suggestions for Authors
The manuscript can be accepted in its current version as the author has revised it according to the reviewers' comments.
Author Response
Thank you for your approval.
Reviewer 3 Report
Comments and Suggestions for Authors
I am satisfied with the response provided by the authors.
Author Response
Thank you for your approval.